# Potential Properties of Natural Nutraceuticals and Antioxidants in Age-Related Eye Disorders

**DOI:** 10.3390/life13010077

**Published:** 2022-12-27

**Authors:** Jessica Maiuolo, Rosa Maria Bulotta, Francesca Oppedisano, Francesca Bosco, Federica Scarano, Saverio Nucera, Lorenza Guarnieri, Stefano Ruga, Roberta Macri, Rosamaria Caminiti, Vincenzo Musolino, Micaela Gliozzi, Cristina Carresi, Antonio Cardamone, Annarita Coppoletta, Martina Nicita, Adriano Carnevali, Vincenzo Scorcia, Vincenzo Mollace

**Affiliations:** 1Laboratoy of Pharmaceutical Biology, IRC-FSH Center, Department of Health Sciences, University “Magna Græcia” of Catanzaro, Germaneto, 88100 Catanzaro, Italy; 2IRC-FSH Center, Department of Health Sciences, University “Magna Græcia” of Catanzaro, Germaneto, 88100 Catanzaro, Italy; 3Nutramed S.c.a.r.l, Roccelletta di Borgia, 88021 Catanzaro, Italy; 4Department of Medical and Surgical Sciences, University “Magna Græcia” of Catanzaro, 88100 Catanzaro, Italy

**Keywords:** eyes diseases, oxidative damage, antioxidant compounds, vitamin A, vitamin C, lutein, trehalose, astaxanthin, curcumin, quercetin, Coenzyme Q10, PUFAs, BPF, grape seed

## Abstract

Eye health is crucial, and the onset of diseases can reduce vision and affect the quality of life of patients. The main causes of progressive and irreversible vision loss include various pathologies, such as cataracts, ocular atrophy, corneal opacity, age-related macular degeneration, uncorrected refractive error, posterior capsular opacification, uveitis, glaucoma, diabetic retinopathy, retinal detachment, undetermined disease and other disorders involving oxidative stress and inflammation. The eyes are constantly exposed to the external environment and, for this reason, must be protected from damage from the outside. Many drugs, including cortisonics and antinflammatory drugs have widely been used to counteract eye disorders. However, recent advances have been obtained via supplementation with natural antioxidants and nutraceuticals for patients. In particular, evidence has accumulated that polyphenols (mostly deriving from Citrus Bergamia) represent a reliable source of antioxidants able to counteract oxidative stress accompanying early stages of eye diseases. Luteolin in particular has been found to protect photoreceptors, thereby improving vision in many disease states. Moreover, a consistent anti-inflammatory response was found to occur when curcumin is used alone or in combination with other nutraceuticals. Additionally, Coenzyme Q10 has been demonstrated to produce a consistent effect in reducing ocular pressure, thereby leading to protection in patients undergoing glaucoma. Finally, both grape seed extract, rich in anthocyanosides, and polynsatured fatty acids seem to contribute to the prevention of retinal disorders. Thus, a combination of nutraceuticals and antioxidants may represent the right solution for a multi-action activity in eye protection, in association with current drug therapies, and this will be of potential interest in early stages of eye disorders.

## 1. Introduction

To date, it is known that improving eye health is one of the objectives of the UN Summit on Sustainable Development, and this can be achieved by including promotion, prevention, care, and rehabilitation strategies. Improving eye health includes not only best vision, but also reducing disability and increasing wellbeing [1]. Therefore, it is possible to improve eye health and solve world problems associated with it; for example, the reduction of hunger reduces eye diseases related to malnutrition, but also, better eye health reduces poverty and thus reduces hunger [2]. The normal human eye measures approximately 22 to 27 mm in the anteroposterior diameter and possesses a circumference from 69 to 85 mm. The human eyeball consists of three primary sections: (1) the outer layer of support of the eye, which includes clear cornea, opaque sclera, and their interdigitation, designated as limbo; (2) the central uveal layer of the eye comprising the iris, ciliary body, and choroid; (3) the inner layer of the eye, commonly referred to as the retina [3,4]. The visual process begins with the crossing of light through the clear cornea, the pupillary opening, the crystalline lens, and the retina. Finally, the visual impulse is transmitted to the brain through the optic nerve [5,6]. Light is a form of electromagnetic energy that enters our eyes through the pupil; light converges through the cornea and the lens upon the receptors of the retina that are located on the back wall of the eye. The pupil is surrounded by a pigmented iris that can expand or contract, making the pupil larger or smaller as the incident light level changes. Retinal receptors detect light energy and, through a transduction process, generate the action potentials that then travel along the optic nerve [7,8]. In Figure 1, eye structure and the Table 1 mechanism of vision are shown.

The eye, regardless of its complex structure, is connected structurally and functionally with vascular, nervous, endocrine, and immune tissue. Nevertheless, it is constantly exposed to the external environment, and for this reason, it must be protected from damage from the outside. A very important protection to the eye is guaranteed by the production of a tear film, consisting of aqueous components, electrolytes, lipids, and mucus. In this way, the lubrication is maintained, and any factor that disturbs its composition can alter the stability, osmolarity, tissue structure, and mechanical and inflammatory mechanisms of the eye [9,10,11]. Eye diseases are due to aging, the occurrence of local and specific diseases, or concomitant pathologies. Impaired vision negatively affects quality of life and daily activities, increases the risk of disability and depression in older age, and may lead to loss of independence [12,13,14]. Common eye diseases are ocular atrophy, corneal opacity, cataracts, uncorrected refractive error, posterior capsular opacification, uveitis, glaucoma, age-related macular degeneration, diabetic retinopathy, retinal detachment, and undetermined disease [15,16]. The drug therapy adopted for these pathologies can be supported by the intake of particular foods or food supplements, which have been shown to address the pathophysiologic mechanisms directly involved. Numerous micronutrients and nutraceuticals products, in fact, can affect some eye components through the involvement of different metabolic pathways [17,18,19]. Since many eye diseases have oxidative stress as a common denominator, it is essential to study the main natural antioxidant compounds in order to justify their use in clinical practice. The bibliography used to write this review was downloaded from PubMed using terms “ocular diseases”; “antioxiant and ocular diseases”; and “pharmacology of ocular hepatologies”. This review can be divided into two sections: in the first part, the close correlation between the main pathologies affecting the eye and oxidative damage is described; in the second part, the beneficial effects of some antioxidant compounds on eye health are developed.

## 2. Reactive Oxygen Species and Eye Health

Oxygen (O_2_) is essential for life as it is crucial for aerobic breathing of cells and tissues, as well as for ensuring the energy cycle of life. Under normal conditions, aerobic metabolism predicts that O_2_ undergoes a reduction reaction, giving rise to water (H_2_O). On the contrary, in some situations, there is an incomplete reduction of O_2_, leading to the formation of very unstable and reactive species, known as reactive oxygen species (ROS), and their accumulation [20]. In metabolic reactions, the first ROS to form is the superoxide anion (O^2−^), which is the most abundant of ROS and possesses a half-life of milliseconds. Because of its very short half-life, this radical ion is not able to attack biological macromolecules, but may stimulate chain reactions resulting in the formation of high concentrations of ROS. The O^2−^ is also used by the immune system to kill pathogenic microorganisms; phagocytes produce O^2−^ in large quantities through the enzyme NADPH oxidase and use it to eliminate pathogens through an oxygen-dependent mechanism. Due to the toxicity of O^2−^, all organisms have developed superoxide antagonist enzyme isoforms; superoxide dismutase (SOD) is capable of catalyzing superoxide neutralization, producing hydrogen peroxide (H_2_O_2_) [21]. H_2_O_2_ is the second most-present ROS; it is a small molecule and a non-radical oxygen species, which spreads easily in biological membranes, propagating its effects even at a distance. In addition, H_2_O_2_ can generate other ROS and decompose, converting to water and oxygen through an exothermic reaction. Since the oxidation state of oxygen in H_2_O_2_ is −1 (intermediate between states 0 and −2), H_2_O_2_ can function as both and oxidizing and reducing agent. Reactions in which H_2_O_2_ is cycled as an oxidant is favored in an acidic environment, while reactions in which it behaves as a reducing agent are favored in a basic environment [22]. The hydroxyl radical (OH•) is the most powerful and harmful of all oxygen species, since, despite having a very short half-life, it shows a high reactivity. Therefore, it is a very dangerous compound for the body because cannot be eliminated by an enzymatic reaction and because it reacts with every oxidizable available compound. OH• can damage all kinds of biological macromolecules, and the only means to protect cellular structures is the use of antioxidants or another efficient repair system [23]. Biological macromolecules (lipids, proteins, and nucleic acids) can be damaged by ROS, although lipids are more susceptible to oxidation. There are defensive mechanisms, both exogenous and exogenous, against oxidative attack; if the production of ROS and the ability of antioxidant biological systems to counteract the effects of ROS metabolites are unbalanced, oxidative stress occurs, which produces varying degrees of damage and abnormality in cells [24]. Exogenous antioxidant enzymes include superoxide dismutase (SOD), catalase (CAT), and glutathione peroxidase (GSH-Px) [25]. Oxidative stress involves the onset of many pathologies, including cancer, neurodegenerative diseases, atherosclerosis, cardio-circulatory pathologies, asthma, infertility, aging, dermatitis, hypertension, diabetes, and rheumatoid arthritis, among others [26,27,28,29,30]. Oxidative stress is also involved in eye diseases, such as dry-eye disorder, cataracts, glaucoma, eye surface disorders, retinitis pigmentosa, diabetic retinopathy, uveitis, age-related macular degeneration and toxic neuropathies [31,32]. The human eye is constantly exposed to sunlight, artificial light, and high metabolic activity; in addition, exposure to ionizing radiation, environmental toxins, and chemotherapy contribute to oxidative damage in eye tissues, making the eye particularly susceptible to oxidative damage. Prolonged exposure to ROS, therefore, constitutes a considerable risk for the health of the eye, whose cells, following the damage of their macromolecules, highlight the impairment of the metabolism, resulting in necrotic or apoptotic death [33]. Eye dryness disorder is defined by the dysfunction of the eye surface, which consists in a deterioration of the tear film and which leads to dryness of the eyes. This disorder involves several portions of the eye, including the eyelids, tear glands, and various tissues of the eye surface [34]. A significant increase in oxidative activity, associated with a decrease in antioxidant defenses in the fluids and tissues of the eye, can damage the eye surface, causing serious alteration to the tissues of the cornea and conjunctiva, evolving in the development of eye dryness and visual damage. In fact, in patients suffering from dry eye, in addition to an imbalance in the state of the tear film, the overexpression of the production of ROS on the eye surface was detected [35,36]. Retinal eye diseases are numerous and particularly linked to oxidative stress for three important reasons: 1) the retina is constantly subjected to the action of visible light that generates photooxidation; 2) the work carried out by the retina is considerable and requires a large consumption of oxygen which, during mitochondrial respiration, could favour the accumulation of ROS; 3) PUFAs are particularly present in the retina which is, consequently, susceptible to lipid peroxidation [37]. Diabetic retinopathy is considered, to date, the main cause of blindness in developed countries. Impaired blood glucose levels cause significant damage to vision, which is aggravated in the presence of ROS accumulation. Free radicals damage cells by acting on biological macromolecules; they disaggregate amino acids by altering the protein structure, fragment nucleic acids by damaging DNA, and modify lipids and membrane structure [38]. These alterations lead, over time, to cell death by necrosis. Finally, ROS accumulation damage in diabetic patients causes systemic alterations caused by vascular dysfunction [39,40]. Glaucoma includes a group of disorders in which selective retinal ganglion cell (RGC) loss occurs; RGCs are located on the inner surface of the retina and connect its axons to the optic nerve. This disease is the second cause of blindness in the world and is a progressive optic neuropathy caused, in most cases, by elevated intraocular pressure [41]. Glaucoma is closely related to oxidative stress, and evidence of this correlation has been provided in both animal and human experiments [42]. In the models used, glaucoma was induced by increased intraocular pressure, optic nerve compression, axotomy, reduced blood supply to the optic nerve, and autoimmune damage. In all these induced situations, an increase in ROS, a reduction in antioxidants, and an increase in retinal lipid peroxidation have consistently been found [43]. The mechanism of action involves ROS and biological macromolecules, proteins, and nucleic acids [44,45]. Cataracts consists in the opacification of the lens, which is located inside the eye and has the purpose of filtering the light that penetrates through the pupil, directing it towards the retina [46]. The triggering causes of cataract onset are multiple; including aging, genetic inheritance, metabolic, and environmental or nutritional insults; sometimes, it can be a consequence of other eye diseases (retinal degenerative, uveitis, glaucoma) or systemic diseases such as diabetes [47]. Nowadays, it is increasingly believed that cataracts are caused by the presence of high-molecular-weight protein aggregates, or by the breakdown of the microarchitecture of the lens. This hypothesis involves post-translational modifications of lens proteins that alter their conformation, leading to destabilization and eventual aggregation [48]. Membrane, luminal, or secretory proteins are synthesized in the rough endoplasmic reticulum (ER) and transported into its lumen [49,50,51]. When internal or external factors intervene, the proteins, inside ER, conform incorrectly, becoming misfolded proteins. Poorly folded proteins in the ER trigger cataract formation processes [48]. In cells, misfolded proteins are eliminated or corrected in suitable conformations by specific pathways, including the unfolded-protein response (UPR). However, chronic UPR further amplifies the degradation, modification, and aggregation of the proteins of the lens in a downstream cascade. UPR culminates in calcium imbalance, protein degradation, oxidative insults, redox state disturbances, and loss of antioxidant defense mechanisms [52]. Recent studies have shown that, in patients with cataracts, ROS induced a reduction in DNA methylation in the *Keap1* promoter gene, activating the expression of Keap1 protein. In physiological conditions, Keap1 can increase the nuclear factor and erythroid 2-related factor 2 (Nrf2) proteasomal degradation. Since Nrf2 controls the basal and induced expression of a number of antioxidant-response genes, it is evident that the reduction of Keap1-induced ROS increases the degradation of Nrf2 and, as a result, reduces a cell’s antioxidant control system [53,54,55]. The consequence is that oxidative damage increases exponentially, along with the development of cataracts of the lens. Finally, an excessive production of free radicals alters the redox state, modulating an inflammatory response, which leads to the exacerbation of oxidative damage favoring numerous pathological states [56]. The main negative function of ROS occurs on biological macromolecules, generating lipid peroxidation, DNA oxidation, and protein alteration, which contributes to the damage of cell structure and function. A prolonged condition of ROS-induced cell damage results in the onset of inflammation and pathologies affecting different body districts. Figure 2 shows how the accumulation of ROS can generate eye pathologies.

## 3. Role of Antioxidants and Nutraceuticals in Maintaining Eye Health

As oxidation leads to the onset of damage and in several districts, the role of antioxidants has become of massive interest for doctors and patients in the treatment and prevention of diseases. An antioxidant compound can be defined as a substance that can delay or prevent oxidation [57]. The body uses different strategies against the production and accumulation of ROS: firstly, antioxidant enzymes are used, as already reported, such as CAT, SOD, GSH-px. It is important to remember that some antioxidant enzymes need micronutrients to function properly, such as zinc, selenium, copper and manganese [58]. Secondly, ROS can be reduced or neutralized by the intake of antioxidant nutrients, such as vitamin E (a-tocopherol), beta-carotene, and vitamin C, among others [59]. An insufficient intake of foods with antioxidant function or an unbalanced diet can alter the body’s natural antioxidant system and facilitate the damage induced by ROS. Additional defense mechanisms include antioxidant compounds, such as metallothionein, melanin, and glutathione [60,61]. In the following sections, we will investigate some natural compounds with marked antioxidant activity. Among them, those that are most used in clinical practice and that, to date, are part of the antioxidant mixtures used in clinical trials, have been selected.

### 3.1. Vitamin A and Lutein 

Among dietary factors, vitamin intake has become increasingly positive in reducing oxidative stress and, as a result, improving the outcome of many eye pathologies. Vitamin A is a fat-soluble life-essential group of compounds characterized by an unsaturated isoprenoid chain structure. Famously, vitamin A performs important functions including cell proliferation, reproduction, foetal growth and development, vision in darkness, corneal and conjunctiva development, immune system functioning, and central nervous system formation [62]. Unlike water-soluble vitamins, vitamin A easily accumulates in the body, especially in the liver and adipose tissue. This characteristic determines the advantage of resisting the development of clinical deficiency symptoms, but at the same time, it has the disadvantage of being able to provide accumulation toxicity. Vitamin A can be supplied in the diet both from products of animal origin, such as retinol, and vegetable carotenoids, such as provitamin A [63]. As already discussed, the retina is responsible for visual perception, mediated through specific structures known as cones and rods that are fundamental for vision in light and dark conditions, respectively. Active vitamin A is associated with a protein receptor coupled to protein G, the complex known as rhodopsin, generating a chain of reactions whose last consequence is the transmission of optical perceptions to the brain via the optic nerve [62]. Vitamin A is responsible for the maintenance of homeostasis reduction–oxidation. In fact, retinol binds to different proteins, acting as a redox reagent. In addition, carotenoids, such as β-carotene, α-carotene, lutein, lycopene, and cryptoxanthin, are well-known antioxidants [64,65]. An excessive intake of vitamin A is potentially toxic; toxicity has been associated with 100,000 RE per day (1 RE = 1 μg retinol) in adults and 10,000 RE per day in children. In women in the first trimester of pregnancy, a dose of 3000–9000 RE per day created teratogenic effects. The Group of Experts on Vitamins and Minerals have not been able to set a safe limit for Vitamin A; therefore, it is recommended that a maximum intake per day is 700 lg for men and 600 lg for women [66]. Among the approximately 850 types of naturally occurring carotenoids, very few of them are present in human tissues. Among these it is important to mention lutein and its stereoisomers, zeaxanthin and meso-zeaxanthin, present in the human retina [67]. It is a class of a carotenoid, named xanthophyll, that cannot be synthesized de novo in the human body, and are absorbed only through diet. The most xanthophyll-rich foods, including lutein and zeaxanthin, are leafy green vegetables, such as cabbage, broccoli, peas, spinach, lettuce, and also egg yolk [68]. These compounds are hydrophobic; however, due to the presence of the hydroxyl group, lutein and zeaxanthin are relatively polar compounds. After their intake, food carotenoids are dispersed in gastric juice and incorporated into lipid droplets, transferred to micelles with food lipids, and finally into the bloodstream. Therefore, fat-rich diets generally facilitate the absorption of dietary carotenoids [69]. Lutein is particularly concentrated in the central portion of the retina, where the photoreceptor cells, responsible for visual acuity and central vision, are located. In the macula of the retina are found: zeaxanthin, in the middle-peripheral region; meso-zeaxanthin, in the epicentre; and lutein, concentrated in the periphery. The absence of these three cartenoids is often used to predict the risk of developing macular diseases [69]. Lutein is also found in the lens, protecting it from age-related eye diseases, such as cataracts [70]. Lutein is retained in the human retina for a prolonged period of time and it has been shown that, even after three months from the interruption of lutein supplementation, the optical density of the macular pigment remains high despite its low serum concentration [71,72]. Lutein has been shown to exert an extremely powerful antioxidant action with several mechanisms of action: (1) render the role of the oxygen singlet poorly active; (2) reduce or eliminate free radicals; (3) filter blue light, thereby reducing phototoxic damage to photoreceptor cells; (4) reduce the expression of inducible nitric oxide synthase (iNOS) [73,74,75]. In addition, lutein is able to turn off the inflammatory process, inhibiting the pro-inflammatory cytokine cascade, the expression of nuclear-kB transcription factor (NF-kB), and the activation of the complement system [76,77,78,79]. Many clinical studies have attributed lutein to antioxidant and anti-inflammatory properties in the eye, justifying a benefit in some diseases such as age-related macular degeneration, diabetic retinopathy, cataracts, retinitis pigmentosa, and myopia [80,81]. Lutein supplementation showed a relatively high safety profile and was classified as “GRAS” by the US Food and Drug Administration (FDA) [82]. Although German, Canadian and American studies have reported daily intakes of lutein of 1.9, 1.4, and 2 mg, respectively [83,84], these results have been shown to be undervalued, and the randomized clinical trial “Age-Related Eye Disease Study 2” (AREDS 2) showed a lutein intake of 10 mg per day over 5 years in more than 4000 patients. Subsequently, the Council for Responsible Nutrition (CRN) stated that lutein intake is safe up to 20 mg/day [85].

### 3.2. Vitamin C and Coenzyme Q10

Vitamin C (ascorbic acid) is, chemically, a low-molecular-weight carbohydrate capable of donating electrons to free radicals from both the second and third carbon, quenching their reactivity and acting as a reducing agent. While most vertebrates can synthesize this compound, humans, together with guinea pigs, some fish, birds, and insects, rely exclusively on dietary intake to maintain the body’s levels of vitamin C. This is a hydrophilic vitamin, and despite its small size, it does not cross the plasma membrane by passive diffusion [86]. During the detoxification reactions of ROS, vitamin C oxidizes to dehydroascorbate, but this oxidized form can subsequently be reduced to generate vitamin C again by glutathione-dependent enzymes. However, if the oxidative damage is continuous, dehydroascorbate undergoes an irreversible degradation; in the event in which vitamin C is present in excessive doses, it can act as a pro-oxidant, contributing to the formation of hydroxyl radicals and increasing oxidative damage. This means that vitamin C can pass from being an antioxidant in physiological conditions to a pro-oxidant under pathological conditions [87]. Vitamin C is particularly present in the eye in the aqueous humor (saline fluid that is located between the cornea and the crystalline) and vitreous humor (connective tissue of gelatinous consistency occupying the eyeball cavity between the posterior surface of the lens and the retina). Particularly in these districts, its concentration exceeds plasma concentrations by 20 to 70 times [88]. In the eye, vitamin C absorbs UV light, preventing the penetration of UV rays and subsequent photoinduced oxidative damage in the tissues, behaving as a physiological “sunscreen” [89]. In addition, vitamin C can scavenge or quench the superoxide anion radical, hydrogen peroxide, hydroxyl radical, singlet oxygen, and reactive nitrogen oxide [90], protecting the cornea, the lens and other ocular tissues against oxidative damage. Finally, vitamin C has been shown to play a role in the prevention of lipid peroxidation of membranes [91]. Since it has been shown that individuals with vitamin C deficiency developed cataracts more easily than others, and that there was a close correlation between vitamin C and the health of the crystalline, numerous studies have been conducted on the relationship between vitamin C and the risk of cataracts [92,93]. The recommended daily allowance (RDA) based on the intake of vitamin C, is 75 and 90 mg/day for women and men, respectively, established by the U.S. Institute of Medicine (IOM) in 2000 [94]. Recent data suggest that the current RDA for vitamin C set by the IOM for men and women may be too low. On the basis of a comprehensive review of the scientific evidence, it was concluded that 200 mg/d is the optimum intake of vitamin C for the majority of the adult population, to the advantage of the health benefits for the eye [95]. Coenzyme Q10 (coQ10) possesses a quinone structure and, for this reason, is also known as ubiquinone. The chemical structure of coQ10 is very similar to that of vitamin K; nevertheless, this cofactor is not considered a vitamin because it is the only fat-soluble antioxidant that animal cells synthesize de novo in the body [96]. It is found in all cell membranes and its main function is to be a cofactor of the mitochondrial enzymes that cooperate in the formation of ATP, an energy source needed to perform cellular biochemical functions. In particular, this liposoluble compound works to transport the electrons in mitochondria during aerobic cellular respiration, from complex I (NAHD ubiquinone oxioreductase) and complex II (succinate ubiquinone reductase) to complex III (ubiquinone cytochrome c reductase). Another function of coQ10 is to participate in the creation of a proton gradient in the intermembrane space [97]. This compound also possesses direct and indirect antioxidant properties in its reduced form (CoQ10H_2_). The direct antioxidant property is achieved by reducing the accumulation of ROS, while the indirect action occurs with the regeneration of a form of vitamin E (α-tocopherol) [98]. CoQ10 collaborate in lowering the lysosomal pH, transporting H^+^ ions inside, in order to facilitate an acidic environment necessary to degrade cellular debris [99]. Finally, it has been recognized that coQ10 participates in gene expression, and this could explain its effects on overall tissue metabolism [100,101]. Since the quantity of coQ10 present in the body is determined by two sources, biosynthesis [102,103] and dietary supplementation, its deficiency may occur for the following reasons: (1) reduced dietary intake; (2) impaired biosynthesis; (3) increased usage by the body [104]. A shortage of coQ10 is mainly manifested by reduced energy metabolism, impaired protection from free radicals, and deacidification of lysosomes [105,106,107]. Since the retina is the most metabolically active tissue of the body, with the highest consumption of energy (tissue/size ratio), patients with coQ10 deficiency may develop retinopathies, suggesting that coQ10 can play an important role in pathogenesis of retinal conditions [108,109]. In addition, a study by Que et al. showed higher coQ10 concentrations in young people (30 years) compared to older human retinas (80 years), highlighting how the oxidative stress plays a key role in the pathogenesis of many age-related diseases, such as atherosclerosis, cataracts, and Alzheimer’s disease [110,111,112]. In this way, the accumulation of ROS in aging results in increased cell damage that mediates the apoptotic mechanisms of cell death [113]. Age-related macular degeneration (AMD) causes loss of central vision, which has a significant impact on quality of life. Plasma coQ10 levels are substantially reduced in patients with AMD, compared with control patients, and this suggests an association between coQ10 and AMD pathogenesis [114]. Since the retina and the ocular macula are exposed to light more than any other organ or tissue in the body, these districts will be particularly sensitive to oxidative stress and lipid peroxidation. The result of this oxidative damage leads to apoptotic cell death [115]. Glaucoma is characterized by the loss of RGC, which are fundamental in the transmission of the signal from the photoreceptors to the optic nerve. As the prevalence of glaucoma increases with age, there may be a possible correlation between RGC and coQ10 deficiency in old age [116,117]. Experimental studies have shown that intravitreal administration of coQ10 minimizes apoptosis in RGC. This supported the neuroprotective role of coQ10 [118,119]. The normal concentration range of coQ10 in human plasma is 0.8–1.2 mg/L, and in cases of deficiency, supplementation typically given to adults at 1.2–3 g/day [120].

### 3.3. Astaxanthin 

Carotenoids are a class of compounds with over 600 natural fat-soluble pigments that play a crucial role in the photosynthetic process, as well as protective activity against damage caused by excessive exposure of light and oxygen. They are taken through the diet, constituting a robust nutritional role as a source of vitamin A [121]. Their oxygenated derivatives are known as xanthophylls; both classes of pigments share a structural scheme consisting of alternating single and double bonds, responsible for the absorption of the excess energy contained in other molecules and carrying out, accordingly, an antioxidant role [122]. Astaxanthin, a xanthophyll carotenoid, is a naturally occurring red pigment in numerous living organisms (bacteria, microalgae and yeasts), present as a secondary metabolite. This compound is biosynthesized by phytoplankton and microalgae (such as *Haematococcus pluvialis, Chlorella zofingiensis* and *Xanthophyllomyces dendrorhous),* which are the basis for the feeding of zooplankton and krill, the ideal food of animal species which store this pigment in the skin and adipose tissue. Finally, all the superior aquatic species that feed on these foods become rings of the trophic chain containing astaxanthin. This carotenoid is responsible for the colouring of some sea creatures, including salmonids, tuna, shrimps, crustaceans, lobsters, and crayfish [123]. The chemical structure of astaxanthin provides a long chain structure and two terminal polar groups; this particular conformation provides the compound with both lipophilic and hydrophilic properties. Precisely because of this feature, astaxanthin extends through the entire double-layer membrane, carrying out its protective activity both inside and outside the cell membrane [124,125]. Astaxanthin has been shown to have numerous beneficial activities on human health, including a protective effect on the skin, the cardiovascular and nervous system, and antioxidant, anti-inflammatory, anti-cancer, and antidiabetic properties, among others [126,127]. In the last two decades, the potential role of astaxanthin in protecting eye health has been highlighted, showing a significant improvement in macular degeneration, cataracts, diabetic retinopathy, and glaucoma [128]. To date, astaxanthin is considered to be the most beneficial antioxidant carotenoid provided in nature; in fact, it is more potent than most known antioxidants, according to the following ratios: 6000 times more powerful than vitamin C; 550 times stronger than vitamin E (alpha-tocopherol); and 40 times more powerful than beta-carotene [129]. The reason for this exceptional oxidative protection is explained by its chemical structure: the terminal polar groups quench free radicals and the double bonds, its intermediate segment, removes high-energy electrons [130]. The peculiarity of astaxanthin is to be able to neutralize single oxygens and radicals in both the non-polar (hydrophobic) and polar (hydrophilic) zones [131]. Several studies have confirmed the antioxidant efficacy of astaxanthin and have identified the reduction of oxidative marker levels, such as malondialdehyde (MDA), and the increase in antioxidant agents including SOD, CAT and GPX1 [132,133,134]. Therefore, we can conclude that astaxanthin exerts antioxidant activities not only through direct scavenging of radicals, but also by activating the cellular antioxidant defense system [135,136]. Finally, astaxanthin exerts a robust anti-inflammatory [137,138] and anti-apoptotic activity [139,140]. In the last two decades, it has been shown that treatment with astaxanthin improves many eye conditions. For example, Otzuka et al., showed that treatment with axanthin 100 mg/kg could inhibit retinal dysfunction caused by light [141]. At the same time, Parisi et al. pointed out that patients with nonadvanced AMD could improve their clinical condition. In particular, a study compared a group of 15 patients who were treated with oral supplementation of astaxanthin (4 mg), zeaxanthin (1 mg), lutein (10 mg), vitamin C (180 mg), vitamin E (30 mg), zinc (22.5 mg), and copper (1 mg) daily for 12 months with a control group (12 patients). The results showed that patients treated with supplementation showed selective improvement of retinal function, compared with the control group [142]. The administration of astaxanthin/lutein/zeaxanthin over a two-year period has been able to improve visual acuity, contrast sensitivity, and vision-related functions [143]. Diabetic retinopathy, the main complication of diabetes, is considered to be the result of chronic oxidative stress and inflammation. The bioactive compound astaxanthin has been shown to exert neuroprotective effects in experimental models of diabetic retinopathy, reducing oxidative stress, inhibiting the activity of NF-κB transcription factor, and reducing the expression of inflammation mediators [144,145]. To confirm this, Yeh et al. demonstrated in a model of rats with streptozotocin-induced diabetes that treatment with astaxanthin resulted in a reduction in histological lesions typical of diabetic retinopathy, oxidative stress, and inflammation, an increase in antioxidant enzymes and a reduction in the expression of NF-κB [146]. The use of astaxanthin has been shown to reduce lens opacification in cataracts induced by prolonged steroid treatment or hyperglycemia [147,148]. Growing evidence suggests that astaxanthin has numerous beneficial effects in several eye diseases affecting both the anterior and posterior poles. Finally, this compound showed an optimal safety profile with no adverse events in any clinical study [149].

### 3.4. Trehalose

Trehalose, also referred to as α,α-Trehalose or α-D-Glucopyranosyl-α-D-glucopyranooside, is a non-reducing disaccharide consisting of two portions of glucose linked through an α,α1,1-glucosidic bond [150]. This component is usually found in the environment in different species of plants, fungi, algae, bacteria, yeasts, insects, and other lower invertebrates, but never in mammals or other vertebrates. Once ingested with food, trehalose is hydrolyzed in the small intestine in two molecules of D-glucose through the trehalase enzyme. It is also known to be used as a sweetener; in industry as, a stabiliser or packaging material; in the pharmaceutical industry, as an excipient; and in the textile industry, as a texturing agent 2 [151]. The chemical structure of trehalose justifies the impossibility of crossing cell membranes; to date, it has been established that this compound can penetrate the cytosol through vesicular internalization of endocytosis [152,153]. Trehalose has also recently been used for its therapeutic effects in the treatment of cardiometabolic disease and degeneration in animal and human models [154]. Takahashi et al. [155] found that trehalose reduced neuronal damage in a model of ischemia in rabbits. In the case of neurodegeneration or traumatic brain injury, oral administration of 2% trehalose has been shown to improve the cognitive characteristics involved in injured brain areas [156]. In addition, this compound induced the increase of zinc and iron in the brain, essential for the maintenance of brain functions [157,158]. Finally, trehalose increased the expression of proteins involved in synaptic activity, neuroprotection, and neurogenesis [159]. Trehalose has become known for its pro-autophagic action; it is able to eliminate and recycle damaged macromolecules in response to cellular stress. For this reason, it was considered protective against the aggregation of the β-amyloid protein in neuronal cell lines [160] and metabolic disorders. More generally, it is possible to affirm that the action of this compound guarantees homeostatic paths that depend on the type of tissue involved [161]. Trehalose has been found to be a stress-reactive factor; in particular, it is a response factor when the cells of the organisms that produce it are exposed to environmental stress, such as heat, cold, dessication, and oxidation. When stress conditions become excessive, these organisms synthesize trehalose, which helps them maintain cellular integrity. With this strategy, these organisms can prevent the denaturation of proteins, facilitate their stabilization, and inhibit protein aggregation [162,163]. Another important function of trehalose is to properly moisturize the dried tissues of the organisms that synthesize this compound. This strategy, known as the “water hypothesis”, is based on the ability of trehalose to form hydrogen bonds between its polar groups and lipids or membrane proteins [164]. In this way, trehalose can maintain the integrity of phospholipids guaranteeing the functional properties of biological membranes [165]. There is a large volume of evidence that testifies to a specific role of trehalose in performing antioxidant and anti-inflammatory effects and, in recent years, has shown beneficial effects in ophthalmology [166]. For example, cell exposure to H_2_O_2_ causes severe oxidative damage to the amino acids of cellular proteins; nevertheless, trehalose is able to reduce such damage, in this way protecting both proteins and lipids of the membrane [167]. The work produced by Cejková et al. showed a protective effect of trehalose on cornea damaged by ultraviolet radiation B (UVB). In addition, ocular tissues, exposed to photodamage, react more easily by suppressing oxidative, inflammatory, and apoptotic pathways when exposed to this compound [168]. Trehalose has an established ability to maintain the degree of hydration; this compound has been used to protect the cells of the anterior ocular surface in dry-eye disease [169]. In addition, the effect of trehalose was enhanced when it was added to a single formulation with hyaluronate, an anionic glycosaminoglycan polysaccharide with lubricative and water-retaining properties [170]. The trehalose highlighted the ability to reduce photodamage caused by UVB radiation in the epithelium of the cornea [171]. Following UVB irradiation, trehalose was found to reduce the resulting corneal disorders, speeding up healing, suppressing neovascularization, restoring corneal transparency, and restoring proper immunohistochemistry [172]. Chen and Haddad (2004) also highlighted the effectiveness of trehalose against hypoxic or anoxic lesions. The cornea is not only affected by damage caused by ROS, but also by insufficient oxygen supply. In these cases, apoptotic death occurs; trehalose has been shown to effectively suppress these disorders [173].

### 3.5. Curcumin and Quercetin

Curcumin is a pigment insoluble in water extracted from the rhizome of *Curcuma longa*, a species that belongs to the *Zingiberaceae* family. The obtained powder contains 2–5% curcumin. Curcumin is a biologically active phytochemical compound with health benefits, and its extract also contains β-carotene, lycopene, epigallocatechin gallate, and quercetin [174,175]. In recent years, several studies have confirmed the use of curcumin for the prevention and treatment of many diseases, especially inflammatory diseases and cancer [176,177]. Curcumin inhibits the production of free radicals, and therefore shows antioxidant properties [178]. The effect of curcumin on oxidative damage is based on its ability to scavenge ROS and reactive nitrogen species (RNS) [179]. It can modulate the activity of active enzymes in the neutralization of free radicals, GSH-Px, CAT, and SOD [180]. In addition, it can inhibit enzymes that generate ROS, such as lipoxygenase/cyclooxygenase and xanthine hydrogenase/oxidase Finally, being lipophilic, curcumin is able to eliminate peroxylic radicals, behaving like vitmin E. Antioxidant properties of curcumin cause the inhibition of oxidative stress and this reduces the risk for many lifestyle diseases [181]. In addition, curcumin exerts an anti-inflammatory effect; it lowers the expression of the gene IκBα, gene cyclooxygenase-2 (COX-2), prostaglandin E-2 (PGE-2), interleukin 1-6-8 (IL-1, IL-6, IL-8), and tumor necrosis factor α (TNF-α). The anti-inflammatory effect is also exerted by the ability of curcumin to activate the proliferator-activated peroxisome receptor γ (PPAR-γ), a nuclear receptor protein that binds to the peroxisome proliferator response element (PPRE) and regulates gene transcription [182]. Recently it has been highlighted that ROS are fundamental regulators of angiogenesis, the process that allows new blood vessels form within the vascular system, and that vascular function critically depends on the amount of ROS present. In fact, while high doses of ROS induce oxidative stress and subsequent cell death, both conditions inhibit angiogenesis; low doses of ROS promote it through some sublethal damage to the cell membrane, subsequently releasing the growth factor of fibroblasts FGF-2, which are directly involved in angiogenesis [183,184]. Given the properties of curcumin, this compound could be used in the treatment of diseases related to angiogenesis, including eye diseases [185,186]. Most retinal diseases, as already mentioned, imply an abundance of ROS and reduced levels of scavenger antioxidants. In particular, RGC and photoreceptors are extremely sensitive to oxidative stress damage, and it is known that the accumulation of ROS is often involved in several diseases of the retina, such as uveitis, age-related macular degeneration, diabetic retinopathy, retinal tumors, and proliferative vitreoretinopathy (PVR) [187]. In order to know the effect of curcumin after an oxidative stress insult, Munia et al., have shown that this nutraceutical compound was capable of protecting human retinal epithelial cells from death [188]. Dry-eye disease is characterized not only by reduced secretion of tears, but also rapid tear evaporation, responsible for the damage to the eye surface [189]. This disease includes an inflammatory process involving IL-6, IL-8, IL-1β [190]. It has been shown that curcumin could exert a protective effect through its anti-inflammatory activity, inhibiting the expression of pro-inflammatory cytokines in conjunctiva [191]. Uveitis is an inflammation of the eye that includes the iris and adjacent tissue. Lal et al. reported an improvement in patients with chronic uveitis who have been given oral capsules of curcumin (75 mg/capsule) [192]. A decrease in aqueous flare and keratic precipitates was observed after treatment. Its beneficial effects can be derived from their antioxidant, anti-inflammatory, and antifibrinolytic properties [189]. Because curcumin has low oral solubility and bioavailability, its biomedical potential cannot be exploited in animals and humans [193]. The exogenous curcumin administered in humans for a period of about 8 weeks has been shown to be able to adequately perform all the functions of this natural compound [194]. Quercetin is a member of the subclass of flavonols and is abundant in the human diet. It has received considerable attention from the scientific community in recent years thanks to numerous effects on human health, including antioxidant [195], anti-inflammatory [196], anti-cancer [197,198], anti-aging [199], and anti-autoimmune effects, [200] and effects upon metabolic pathologies [201]. The eye surface serves as a protective and functional barrier for the rest of the eye. Diseases of the eye surface can affect the structure of the cornea or conjunctiva, leading to corneal thinning, inflammation, and visual deficits [202]. Prolonged inflammation in these districts can lead to a partial or even complete loss of vision, affecting quality of life. Unfortunately, to date, we do not possess a non-invasive treatment that can preserve corneal function; surgery and corneal transplantation remain the only solution [203]. Given the problems and side effects involved in a corneal transplant, scientists have sought an alternative. The effectiveness of the administration of ocular drugs depends on many factors, including drug absorption, bioavailability, and retention on the front surface. Lipophilic drugs, in general, are associated with increased corneal epithelial permeability, and the solubilization of these compounds in water eye drops was tested. Topical application of quercetin or other flavonoids may be more effective in treating conditions affecting the eye surface [204]. Studies have shown that a protective role is carried out by quercetin when administered at an average daily consumption of about 16–23 mg/day in human populations [205,206,207]. In Figure 3, the chemical structures of the examined compounds are represented.

### 3.6. PUFAs

Polyunsaturated fatty acids (PUFAs) are lipids whose hydrocarbon chain has a polar hydrophilic end with a carboxyl group (-COOH) and another end with a non-polar hydrophobic methyl group (-CH3). The n-3 and the n-6 represent two classes of PUFAs defined as “essential”, as they must be taken with the diet because humans do not have the desaturases Δ12 and Δ15, enzymes that catalyze the formation of double bonds along the hydrocarbon chain [208]. Specifically, linoleic acid (LA, 18:2) is the n-6 PUFA from which γ-linolenic and arachidonic acid derive, while from α-linolenic acid (ALA, 18:3), one derives n-3 PUFAs, such as eicosapentaenoic acid (EPA, 20:5 n-3) and docosahexaenoic acid (DHA, 22:6 n-3). LA is present in safflower, soy, and corn oils, while the vegetables that contain ALA are flaxseed, beans, nuts, and the leaves of some green plants. In the liver, the amount of EPA and DHA obtained from the metabolism of ALA is very small, therefore they must be taken with the diet. Both lean and fatty marine fish, fish oil, and algal-derived supplements are particularly rich in EPA and DHA, although they are present in small quantities in many foods of animal origin. For this reason, they must be present in the daily diet [209]. In particular, EPA and DHA can be present in cellular phospholipid membranes and have powerful antioxidant and anti-inflammatory effects [210]. The positive effects of taking n-3 PUFAs are well documented; in fact, the long-term effects against certain pathologies, such as cardiovascular diseases, neurodegenerative diseases, and osteoarthritis, are known [211]. The n-3 PUFAs are also implicated in diseases affecting the eye. For example, in the case of neovascular eye diseases, such as retinopathy of prematurity, diabetic retinopathy, and age-related macular degeneration, current therapies have significant side effects. Clinical and experimental investigations have shown that such treatments could be accompanied by a higher intake of n-3 PUFAs [212]. Despite conflicting results between fundamental and clinical research, it appears that PUFAs can act positively on the damage that determines the pathogenesis of glaucoma [213]. Indeed, the effect of PUFAs can be both on intraocular pressure (IOP) and on survival of RGCs. In the first case, endogenous prostaglandins (PGs), obtained from the metabolism of PUFAs, by activating the EP4 and FP receptors, reduce IOP. In the second case, thanks to their anti-inflammatory and antioxidant effects, n-3 PUFAs can reduce the inflammation and oxidative stress responsible for the RGC dysfunction or death [214]. In particular, patients with pseudoexfoliative (PEX) glaucoma have benefited from the administration of a high, rich DHA nutraceutical formulation that reduced oxidative stress and inflammation. The fact that PUFAs may have beneficial effects for glaucoma is linked to their positive action on endothelial dysfunction and atherosclerosis [215]. Regarding cataracts, a study was conducted on male Wistar rats, in which it was found that the antioxidant and anti-inflammatory activity of lutein increased in the presence of EPA + DHA [216]. In particular, micellar lutein with EPA + DHA has been shown to positively regulate α-crystalline chaperone function [217]. Chang et al. demonstrated a reduction in free fatty acid levels in patients with senile cataracts compared with normal controls. The levels of DHA were particularly low [168]. Studies on the molecular mechanisms responsible for aging in the eye have allowed us to define the involvement of the ELOVL2 (elongation of very-long-chain fatty acids-like 2) enzyme in the regulation of molecular aging in the retina [218]. The ELOVL2 enzyme catalyzes an elongation reaction of n-3 and n-6 PUFAs, which are essential for retinal function. DHA is the main PUFA in the retina and is involved in the photoreceptor function in retinal development and has an antioxidant role [219]. The involvement of PUFAs in AMD, which is the leading cause of blindness in the elderly, has been demonstrated. In particular, the analysis of the eyes of subjects with AMD showed a reduction in PUFAs, and a direct correlation was also reported between the reduced dietary intake of n-3 PUFAs and the increased risk of AMD. Given the function of the ELOVL2 enzyme, it appears that this is directly involved in the onset of AMD along with PUFAs [220]. Furthermore, it appears that DHA may reduce the risk of AMD occurrence by stimulating the synthesis of endogenous antioxidants and the selective autophagy of misfolded proteins [221]. In a study conducted on the eyes of patients with AMD, the analysis of the lipid profile of the retina reported a low ratio between n-3 and n-6 PUFAs, demonstrating the protective role of dietary n-3 PUFAs against AMD [222]. The efficacy of PUFA treatment has also been studied in patients suffering from a multifactorial inflammatory disease, such as dry-eye disease (DED) [223]. The improvement, over time, of parameters such as tear breakup time (TBUT), ocular surface disease index (OSDI), osmolarity and Schirmer’s test, led to the conclusion that PUFAs have a positive effect on nonspecific typical DED when administered briefly and not in combination with other eye medications. Other authors acknowledge the efficacy of n-3 PUFA supplementation in managing DED, even though they consider the evidence to be uncertain and inconsistent [224]. Numerous studies indicate that PUFAs derived specialized-pro-resolving mediators (SPM) capable of maintaining ocular surface health and immune homeostasis, thanks to the fact that SPM pathways and receptors are highly expressed on the ocular surface [225]. Here, the SPMs, produced endogenously, regulate wound healing, innate immunity, and nerve regeneration. SPMs are involved in the protection of the cornea; they are present in significant quantities in healthy human tears; and they guarantee an anti-inflammatory state by increasing the speed of healing of corneal wounds. SPMs reduce the risk factors for corneal transplant rejection by improving graft viability and inhibiting the initiation of alloimmunity [226]. Furthermore, endogenous SPMs have been shown to improve nerve regeneration in the stroma and cornea, representing potential topical therapies in the case of corneal diabetic neuropathy. Some SPM appear to improve symptoms of allergic conjunctivitis, such as reduced mucin secretion and total conjunctival immune cell count. They can also be used effectively as topical treatments for immune-driven DED, as well as for the reduction of ocular surface damage due to viral or bacterial infections, also caused by the use of contact lenses [227]. Furthermore, variations in the levels of very long-chain (VLC)-PUFAs and in the n-3/n-6 ratio were studied in two different experimental models, such as the spontaneously diabetic Nile rat and the Akita mouse, which represent genetic models of diabetes. The evaluations were made in diabetic conditions and following the integration of n-3 PUFAs in the diet. VLC-PUFAs represent a special class of retinal lipids deriving from PUFAs. The levels of VLC-PUFAs and the n-3/n-6 ratio were also measured in human retinal punches from diabetic and non-diabetic donors [228]. In retinal punches of diabetic and retinopathic patients, the levels of VLC-PUFAs were lower than in healthy subjects of the same age. Dietary supplementation with n-3 PUFAs increased the ratio of n-3/n-6 VLC-PUFAs in both experimental models. Therefore, the authors Gorusupudi et al., concluded that the enrichment of the diet with n-3 PUFAs reduces the risk of diabetes onset and of the retinopathy caused by it [226]. In addition, the effect of dietary supplementation of n-3 PUFAs on myopia, a condition that is increasing worldwide, was investigated in experiments conducted on mice modelling lens-induced myopia (LIM). The results obtained showed that EPA and its metabolites are capable of inhibiting choroidal thinning and myopia progression [229].

### 3.7. Grape Seed Extract and Bergamot Polyphenolic Fraction 

Grape seed extract has shown beneficial effects in many diseases, thanks to its composition as a flavonoid polyphenolic compound; its main components are: +catechin, −epicatechin gallate, gallate and −epigallocatechin [230]. Beneficial properties of grape seed include prevention and treatment of diabetes and its complications [231], prevention of obesity and inflammatory reaction [232,233], the ability to modify early cerebrovascular injury caused by hypertension [234], alleviation of exercise fatigue [235], protection of the myocardium from injury [236], lowering blood lipids, regulating the metabolism, and improving the intestinal flora [237,238]. Most of these activities are carried out thanks to the antioxidant effect of grape seed extract, which makes it more powerful and effective than other plant polyphenolic compounds [239,240]. Among the active ingredients present in grape seed extract, proanthocyanidins are responsible for the biological, therapeutic and pharmacological properties [241]. Many studies have looked at the correlation between grape seed and eye disease: for example, Mani Satyam et al. showed in an in vivo study in rats that the administration of grape seed extract one day before treatment with sodium selenite, capable of inducing cataract formation in animals, was able to significantly reduce the disease and its damage [242]. Another recent study showed that treatment with grape seed extracts was able to reduce the damage of retinal degeneration caused by aging, by attenuating the expression of some pro-inflammatory cytokines [interleukin 6 (IL-6), IL-12 and IL-1β)] or the formation of their messenger RNAs [243]. The protective effects of grape seed proanthocyanidins on retinal ganglion cells have also been confirmed by an important study by Li et al., demonstrating this result in several neurodegenerative disorders [244]. 

Bergamot belongs to the *Rutaceae* family of dicotyledonous angiosperm plants that includes about 1600 species and are characterized by the presence of oleiferous glands producing ethereal aromatic oils. This plant, which belongs to the citrus genus, prefers areas with tropical and subtropical climates, and it prefers a rocky-calcareous soil [245]. The most important citrus fruits are lemon (*Citrus limonum*), sweet orange (*Citrus sinensis*), lime (*Citrus aurantifolia*), mandarin (*Citrus reticulata*), grapefruit (*Citrus vitis*) and bergamot (*Citrus bergamia*). Bergamot is a plant endemic to Calabria, Italy, and its fruit is considered a subspecies of bitter orange or a hybrid derived from bitter orange and lemon [246]. Bergamot pulp and juice are a rich source of polyphenols, including flavonoids, and are used for innumerable human health benefits, such as anti-inflammatory, antiviral [247], antitumor [248,249], hypolipidaemic [250,251], antioxidant, cardioprotective [252], antiplatelet [253] and metabolic protective effects [254]. In addition, bergamot is rich in vitamins, dietary fibre, minerals, and secondary metabolites, including polyphenols, flavonoids, carotenoids, essential oils, sugars, ascorbic acid, and some trace elements [255,256]. Among flavonoids, the main ones are neohesperidin, naringin, neodiosmin, eriodictyol, and neoerythocithrin [257,258]. This composition makes the bergamot’s flavonoid profile unique, justifying its use in many human pathologies [259,260]. Although specific data on the protective effects for the eyes of bergamot polyphenolic fraction (BPF) are not available in the literature, it is hypothetical that the high polyphenolic content and the strong antioxidant properties can also bring benefits to the ocular region. It would be appropriate to carry out relevant trials in order to confirm or deny the potential involvement of BPF on the ocular region.

## 4. Antioxidant Mixtures to Protect/Reduce Eye Diseases

As can be seen from the chemical structure of the compounds examined, the presence of OH groups or the catecholic group explain and justify their antioxidant properties. In fact, data available in the literature indicate that electron donors, particularly the hydroxyl group, are an essential component for performing antioxidant activity [261,262,263,264]. In the last decades, compounds described in this review are not used individually; rather, it is preferred in clinical practice to use mixtures where they are used together. Since their use has not demonstrated adverse effects in a range of concentrations, it is preferable to exploit their synergistic effects. With this strategy, each individual component of the mixture can use its own mechanism of action, providing its support to the common purpose of reducing the severity of eye diseases. Obviously, this type of treatment must be subjected to preliminary scientific studies, which clearly demonstrate that among the components used there is no interference, onset of accumulation effects, or competitive reactions [265,266,267,268,269]. For most eye diseases, the use of a mixture of antioxidants was evaluated, and patients were treated with a specific combination of high doses of zinc and antioxidant vitamins [500 mg of vitamin C, 400 units (about 270 mg) of vitamin E, 15 mg of β-carotene, 80 mg of zinc and 2 mg of copper]. In most cases, this treatment has proved to be efficient, while in healthy patients, the use of the mixture has proved ineffective in preventing the onset of disease [270,271]. Clinical studies have shown the doses of the compounds that should not be exceeded in the different mixtures used. Table 1, generated from literature data, shows the appropriate concentrations for each individual component.

**Table 1 life-13-00077-t001:** Concentration of components to be administered in eye pathology.

Compound	Concentration/Day	Ref.
Vitamin A	0.5 mg	[220]
Vitamin C	300 mg	[221]
Lutein	10 mg	[222]
Curcumin	800 mg	[223]
BPF	250 mg	[254]
Grape seed extract	100 mg	[243]
Quercetin	150 mg	[224]
Coenzyme Q10	100 mg	[225]
Zinc	60 mg	[226]
PUFAs	1–2 mg	[208]

The shown concentration of each component is slightly lower than the recommended daily intake value, so as to avoid potential antagonism or accumulation reactions [272]. The clinical use of antioxidant mixtures is recommended in conjunction with eye disorders in order to avoid exacerbation of the disease [273]. To date, it is common opinion in the clinical field to also administer these mixtures to healthy subjects; this decision should not be made, as already mentioned, in order to prevent the occurrence of eye diseases, but to avoid the accumulation of ROS and the consequent oxidative damage. For this very reason, it would be advisable to use these protective mixtures from the age of 40 years onwards. This can be explained by the need to not already have accumulated excessive amounts of ROS related to aging, to avoid undergoing demanding pharmacological treatments that may affect oxidative stress and have not been subjected to an excess of adverse environmental factors [274,275]. 

## 5. Conclusions

The eye is an organ continuously exposed to ionizing radiation, industrial smoke, pollutants, and engine exhausts, which makes the eyes extremely susceptible to oxidative attack. In addition, it is highly exposed to both light and robust metabolic activity. These two conditions expose the eyes to a continuous phototoxic damage and to an increase/accumulation of oxidative damage, responsible of the principal pathologies that affect their health [56,276,277]. It has been widely shown that sunlight, as well as incandescent lamps, can represent a risk factor for the health of the eye, especially for the retina. To date, the replacement of incandescent light with compact fluorescent lamps (CFLs) or light-emitting diodes (LEDs), is particularly diffuse, and these low-energy devices have increased the possibility of phototoxic eye damage [278,279,280]. On the other hand, oxidative stress, which occurs in the case of an overproduction of ROS or a failure of cellular buffering mechanisms, is able to alter the health of the eye by oxidation of biological macromolecules and an imbalance at the molecular and cellular level [281,282,283]. The main ophthalmic processes in which oxidative stress is involved are: eye surface disorders, glaucoma, diabetic retinopathy, cataracts, toxic neuropathies, uveitis, retinitis pigmentosa, and age-related macular degeneration [284,285,286,287,288]. The human body has some systems of defense against oxidative stress, which are located in the cytoplasm, in the cell membrane, and in the extracellular space: (1) enzyme systems, located in intracellular space, the best known are SOD, CAT and GSH-Px; these enzymes solve the formed ROS. SODs are metalloproteins that accelerate the dismutation of superoxide into hydrogen peroxide. There are two molecular types of SOD in humans: cytosolic (CuZnSOD), which contains copper and zinc, and mitochondrial (MnSOD), which contains manganese [289,290,291,292,293]. CAT is found in peroxisome and mitochondria and its function is to catalyze the dismutation of hydrogen peroxide in water and molecular oxygen [294,295,296,297]. GSH-Px is found in the cytosol and mitochondria. This works by eliminating hydroperoxides, transforming them into water. This reaction is associated with the transformation of reduced glutathione into oxidized GSH [298,299,300]. (2) Free radical scavengers, slow oxidation reactions, or “capture” free radicals transform them into less aggressive compounds. They can be water-soluble, like glutathione and vitamin C, or fat-soluble, like vitamin E and carotenoids. (3) Chelating agents of transition metals are molecules that bind iron and copper, avoiding that these metals act in the reactions of Fenton and Haber-Weiss [301,302,303]. The protective effects of exogenous antioxidants, taken through diet, are added to those generated by endogenous antioxidants, as described. The integration of exogenous antioxidants could take place with the administration of vitamins A, C and E and Coenzyme Q10 [304,305,306]. In this review, the protective roles of some exogenous antioxidants (vitamin A, vitamin C, Coenzyme Q10, lutein, quercetin, PUFAs and curcumin), against the main eye pathologies, in which oxidative damage is directly involved, have been investigated. 

## Figures and Tables

**Figure 1 life-13-00077-f001:**
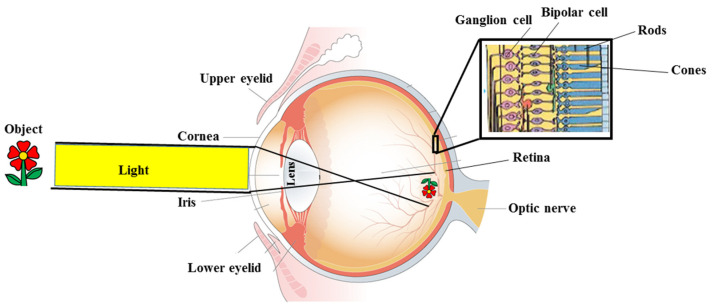
Eye structure and mechanism of vision.

**Figure 2 life-13-00077-f002:**
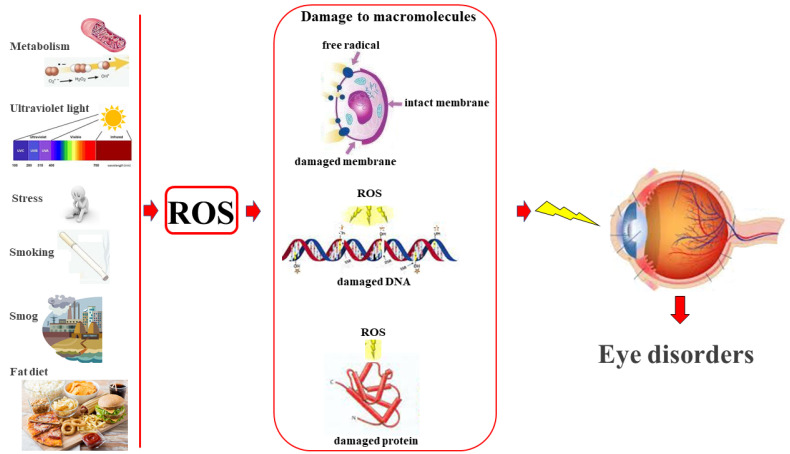
Prolonged damage to biological macromolecules ROS-induced favors the onset of eye diseases.

**Figure 3 life-13-00077-f003:**
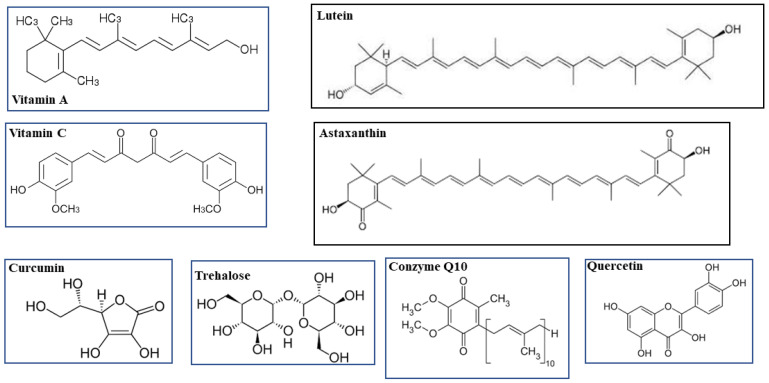
Chemical structures of the examined compounds.

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
