# Peer review of "Potential Properties of Natural Nutraceuticals and Antioxidants in Age-Related Eye Disorders"

_life, 2022, doi:10.3390/life13010077_

Round 1

Reviewer 1 Report

The manuscript “Potential properties of natural nutraceuticals and antioxidants in age-related eye disorders” is a review dealing with a hot topic in the field of eye diseases.

The review is well written and covers most of the antioxidant molecules which have been investigated in the last decades.

However, I would suggest implementing the text with few additional information that can further improve the impact and the significance of this review.

1)    For instance, there are many papers investigating the use of astaxanthin in comparison with other carotenoids. I suggest adding some data

2)    Increasing evidence is provide for the antioxidant properties of trehalose and its use for the treatment of the ocular surface

3)    Finally, I would suggest introducing and additional section focusing on updates on clinical trials on the use of antioxidants for eye diseases.

Author Response

Thank you for your valuable suggestions. The requested changes have been made, as reported in the attached.
Regards,
Jessica Maiuolo

Reviewer 2 Report

Thank you for inviting me to review the manuscript entitled “Potential properties of natural nutraceuticals and antioxidants in age-related eye disorders”. This is an interesting review article collecting and describing the role of nutrition in age-related eye disorders. The manuscript was written well; however, it needs revision prior to publication.

1. the introduction section can be revised to be more straightforward. The justification and rational for conducting this review article should be highlighted.

2. for each nutrient the authors can add a table summarizing available literature with general characteristics of each study. Moreover, results of each study should be shown by an arrow regarding significant or non-significant findings. In addition, clinical and preclinical studies should be reported separately in each section.

3. It would be interesting for readers to find some information about databases used for searching and keywords. Why the authors selected these nutrients?

Author Response

(The authors gave the same response as above.)
